# Cryo-EM structures of the DCPIB-inhibited volume-regulated anion channel LRRC8A in lipid nanodiscs

David M Kern[1,2], SeCheol Oh[3], Richard K Hite[3]*, Stephen G Brohawn[1,2]*

[1]Department of Molecular & Cell Biology, University of California, Berkeley, Berkeley, United States; [2]Helen Wills Neuroscience Institute, University of California, Berkeley, Berkeley, United States; [3]Structural Biology Program, Memorial Sloan Kettering Cancer Center, New York, United States

**Abstract** Hypoosmotic conditions activate volume-regulated anion channels in vertebrate cells. These channels are formed by leucine-rich repeat-containing protein 8 (LRRC8) family members and contain LRRC8A in homo- or hetero-hexameric assemblies. Here, we present single-particle cryo-electron microscopy structures of *Mus musculus* LRRC8A in complex with the inhibitor DCPIB reconstituted in lipid nanodiscs. DCPIB plugs the channel like a cork in a bottle - binding in the extracellular selectivity filter and sterically occluding ion conduction. Constricted and expanded structures reveal coupled dilation of cytoplasmic LRRs and the channel pore, suggesting a mechanism for channel gating by internal stimuli. Conformational and symmetry differences between LRRC8A structures determined in detergent micelles and lipid bilayers related to reorganization of intersubunit lipid binding sites demonstrate a critical role for the membrane in determining channel structure. These results provide insight into LRRC8 gating and inhibition and the role of lipids in the structure of an ionic-strength sensing ion channel.

*For correspondence:
hiter@mskcc.org (RKH);
brohawn@berkeley.edu (SGB)

Competing interests: The authors declare that no competing interests exist.

## Introduction

Volume control is vital for a cell to respond to its environment. Without volume regulation, vertebrate cells are at the mercy of osmotic imbalance, a property utilized at extremes for biochemical cell lysis. However, by exporting or importing osmolytes in response to environmental or intracellular cues, cells can withstand osmotic stresses and actively regulate volume during cell growth, migration, and death.

Vertebrate cells respond to hypotonic environments by opening channels for chloride and other anions, permitting exchange of diverse osmolytes and requisite water across the membrane to alleviate osmotic imbalance (*Jentsch, 2016*). The underlying channels, termed volume-regulated anion channels (VRACs) (*Nilius et al., 1997*) have variable selectivity and conductance between cells, but are similarly activated by osmotic stimuli and blocked by small molecule anionic inhibitors including DCPIB (4-[(2-Butyl-6,7-dichloro-2-cyclopentyl-2,3-dihydro-1-oxo-1H-inden-5-yl)oxy]butanoic acid) (*Decher et al., 2001*). While apparently ubiquitous in vertebrate cells, these channels lacked a molecular identity until 2014, when leucine-rich repeat-containing family member 8A (LRRC8A, also named SWELL1) was identified as a required component of VRAC in cells (*Qiu et al., 2014*; *Voss et al., 2014*). LRRC8A and its paralogs LRRC8B, C, D, and E were proposed to form hexameric ion channels through their homology to Pannexins in the transmembrane helix-containing region (*Abascal and Zardoya, 2012*).

VRACs have been implicated in a wide array of physiological and pathophysiological processes including insulin signaling in adipocytes (*Zhang et al., 2017*), neurotransmitter release from astrocytes and brain damage after stroke (*Hyzinski-García et al., 2014*; *Lutter et al., 2017*), passage of

the chemotherapeutic cisplatin into cancer cells (*Planells-Cases et al., 2015*), and osmotic control during spermatogenesis (*Lück et al., 2018*). A chromosomal translocation in humans that truncates LRRC8A in the LRR region is associated with the B-cell deficiency agammaglobulinemia (*Sawada et al., 2003*). A mouse knockout of *LRRC8A* exhibits increased mortality and developmental defects in addition to significant defects in T cell development and function (*Kumar et al., 2014*). The broad expression of LRRC8s in vertebrate cells suggests VRACs may have additional, yet-undiscovered, roles in cell biology and physiology.

Functional expression of VRAC in cells requires LRRC8A (*Qiu et al., 2014*; *Voss et al., 2014*). LRRC8A can form homomeric channels as well as heteromeric channels with its paralogs LRRC8B, C, D and E; channels have been shown to contain one, two, or three different LRRC8 family members (*Lutter et al., 2017*). Channel properties, including ion selectivity and conductance, are modulated by subunit composition and the diversity of native VRAC properties is presumably a consequence of the different combinations of homomeric and heteromeric channels expressed in different cells. For example, LRRC8A homomeric channels exhibit low conductance relative to heteromeric channels (*Deneka et al., 2018*; *Kasuya et al., 2018*; *Kefauver et al., 2018*; *Syeda et al., 2016*) and only LRRC8AD heteromers readily conduct larger molecules such as cisplatin and the antibiotic blasticidin S (*Lee et al., 2014*; *Planells-Cases et al., 2015*). Mechanistically, LRRC8 channels can be opened by low cytoplasmic ionic strength, one consequence of hypotonic extracellular environments (*Syeda et al., 2016*; *Voets et al., 1999*). However, the precise molecular mechanisms for sensing internal ionic strength and transmitting this stimulus to the opening of a channel gate are unknown.

Here, we report structures of the homohexameric LRRC8A channel embedded in lipid nanodiscs in the presence and absence of the inhibitor DCPIB determined by cryo-electron microscopy. The structures reveal the architecture of the LRRC8 family in a membrane environment, the mechanism of channel inhibition by DCPIB, and provide insight into the role of bound lipids in channel organization. Membrane-embedded LRRC8A channels display significant structural heterogeneity, which we resolve into constricted and expanded states. In both states, the LRR domains are conformationally heterogenous. We propose the differences between constricted and expanded states are related to ionic strength sensing and gating conformational changes. We further compare the structures in lipid nanodiscs to structures in the detergent digitonin presented here and in three recent reports (*Deneka et al., 2018*; *Kasuya et al., 2018*; *Kefauver et al., 2018*). Differences in observed symmetry and LRR conformation between nanodisc-embedded and detergent-solubilized channels suggest an integral role for the membrane environment in LRRC8A structure.

## Results

### Structure of LRRC8A in lipid nanodiscs

LRRC8s are endogenously expressed in vertebrate cells typically used for protein overexpression, so we expressed and purified mouse LRRC8A from *Spodoptera frugiperda* (SF9) insect cells to ensure a homogenous channel preparation for further study (*Figure 1—figure supplement 1*). To assess the activity of channels from this preparation, we reconstituted LRRC8A purified in detergent into phosphatidylcholine lipids and recorded from proteoliposome patches. As is the case in cells and other reconstituted preparations (*Kasuya et al., 2018*; *Syeda et al., 2016*; *Voets et al., 1999*; *Deneka et al., 2018*), LRRC8 channel activity was only observed in low ionic strength solutions (e.g. 70 mM KCl, *Figure 1A*). Channels displayed asymmetric conductance at positive and negative potentials with similar open probability. Purified LRRC8A therefore retains the characteristic properties of homomeric LRRC8A channels in cells: a voltage-dependent conductance activated by low cytoplasmic ionic strength (*Kasuya et al., 2018*; *Syeda et al., 2016*).

Initial electron micrographs of LRRC8A solubilized in detergent displayed heterogeneity between particles, especially in the cytoplasmic LRRs (*Figure 1—figure supplement 2*). Inspired by recent reports of differences between structures of membrane proteins determined in lipidic and detergent environments (*Dang et al., 2017*; *Jin et al., 2017*; *Schuler et al., 2013*), we pursued high-resolution structures of LRRC8A reconstituted in lipid nanodiscs. To this end, LRRC8A was solubilized and purified in detergent, exchanged into lipid nanodiscs formed by the membrane scaffold protein MSP1E3D1 and phosphatidylcholine lipids (POPC, 1-palmitoyl-2-oleoyl-*sn*-glycero-3-

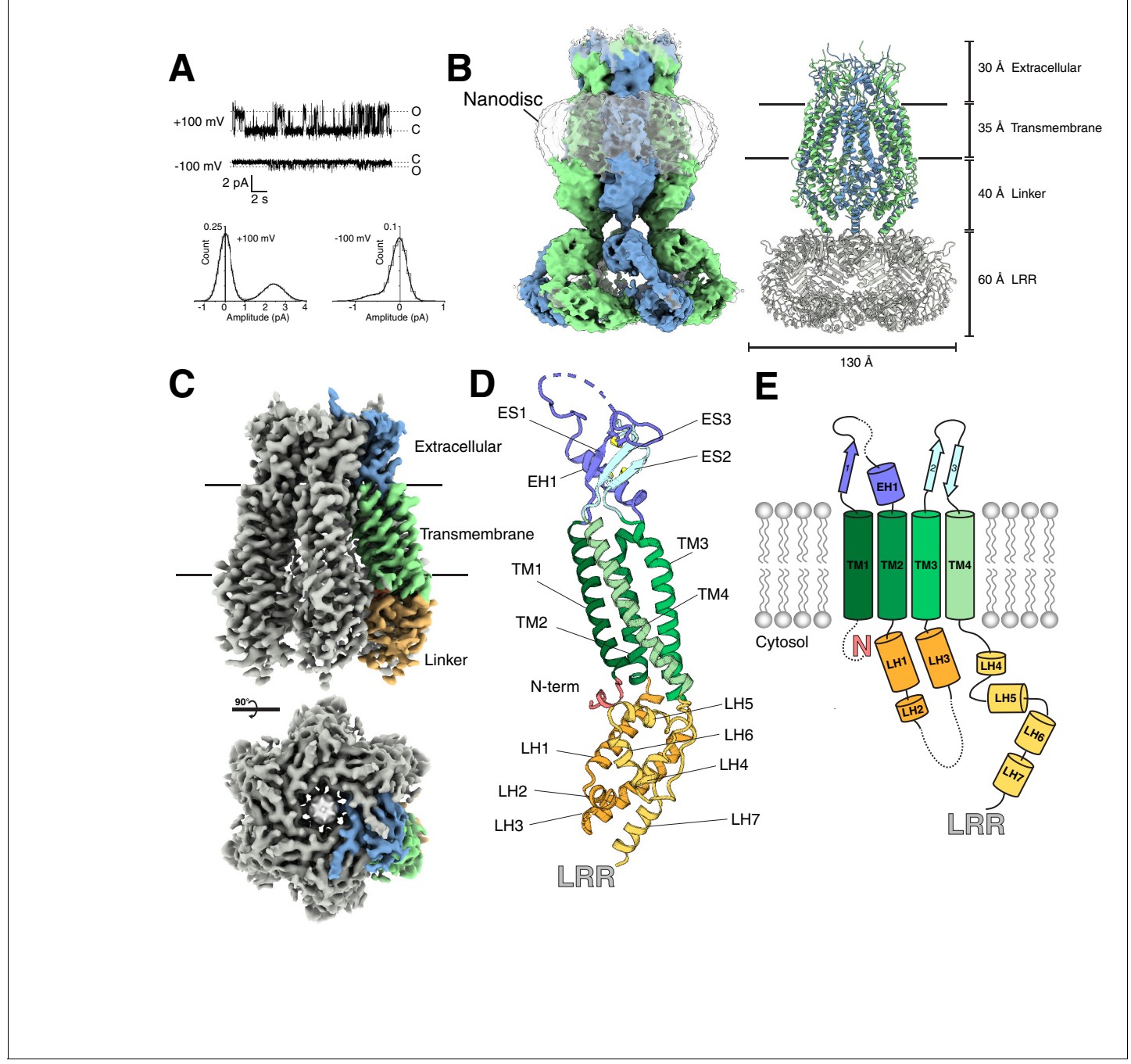

**Figure 1.** Structure of an LRRC8A-DCPIB complex in lipid nanodiscs. (A) Representative single channel recording from an excised patch containing purified LRRC8A reconstituted into phosphatidyl choline lipids ($P_o$ = 0.3, $\gamma$ = 24 pS at +100 mV; $P_o$ = 0.24, $\gamma$ = 5.4 pS at −100 mV). (B, left) Cryo-EM density of LRRC8A from an unmasked refinement of the constricted state at 4.5 Å resolution and (right) corresponding atomic model viewed from the membrane plane. Individual subunits are alternatingly colored blue and green, nanodisc density is rendered transparent, and LRRs docked into the density are colored gray in the model. Dimensions of the extracellular, transmembrane, linker, and LRR regions are indicated. (C) Cryo-EM density of LRRC8A from a LRR masked refinement of the constricted class at 3.21 Å resolution. A view from the membrane (top) and extracellular space (bottom) are shown. One subunit is colored according to region and shown in (C) within the density, (D) as an isolated model with helices and N-terminus labeled and extracellular domain disulfide bonds depicted in yellow, and (E) as a cartoon.

The online version of this article includes the following figure supplement(s) for figure 1:

**Figure supplement 1.** Complex purification and reconstitution.

**Figure supplement 2.** Representative micrographs from LRRC8A preparations in MSP1E3D1 nanodiscs with DCPIB, digitonin, and MSP2N2 nanodiscs.

**Figure supplement 3.** Initial processing for LRRC8A-DCPIB in MSP1E3D1 nanodisc datasets.

*Figure 1 continued on next page*

*Figure 1 continued*

**Figure supplement 4.** The refinement and classification performed to separate constricted and expanded particles for LRRC8A-DCPIB in MSP1E3D1.
**Figure supplement 5.** The final classification and refinement performed on constricted and expanded particles to obtain high-resolution maps for LRRC8A-DCPIB in MSP1E3D1.
**Figure supplement 6.** Constricted map and model validation for LRRC8A-DCPIB in MSP1E3D1.
**Figure supplement 7.** Expanded map and model validation for LRRC8A-DCPIB in MSP1E3D1.
**Figure supplement 8.** LRRC8A in MSP2N2 refinement, particle polishing, and classing into constricted and expanded linker particles.
**Figure supplement 9.** Final particle set 2D classes and the model validation for constricted map of LRRC8 in MSP2N2.
**Figure supplement 10.** Initial processing for the LRRC8A in MSP2N2 nanodisc dataset.

phosphocholine), and vitrified in the presence of the LRRC8 inhibitor DCPIB (*Figure 1—figure supplement 1*) (*Ritchie et al., 2009*).

*Figure 1B* depicts an unmasked reconstruction of LRRC8A-DCPIB in nanodiscs and accompanying model (see *Tables 1* and *2* for cryo-EM data collection and model refinement parameters). LRRC8A forms a 565 kDa, homohexameric channel with each monomer consisting of (from extracellular to intracellular side) an extracellular cap, four transmembrane spanning helices TM1-4, a linker region, and a LRR domain (*Figure 1B–E*). Initial reconstructions showed high-resolution features in the extracellular cap and transmembrane spanning regions, but less well resolved and blurred features in the linkers and LRR regions (*Figure 1—figure supplement 3*). We therefore asked whether distinct conformational states of LRRC8A could be distinguished. Focused classifications on the linker region indeed resolved two channel conformations: a constricted class and an expanded class (*Figure 1—figure supplement 3–5*). Significant structural heterogeneity remains in the LRR region within each class. Masking out the LRRs and applying six-fold (C6) symmetry resulted in the highest resolution reconstructions: 3.21 Å resolution for the constricted (*Figure 1C*) and 3.32 Å resolution for the expanded class, respectively. These reconstructions enabled de novo model building and refinement for all regions except for the LRR domain (*Figure 1* and *Figure 1—figure supplement 6* and *7*), which was instead rigid-body docked into the density from unmasked reconstructions using the high-resolution crystal structure (PDB ID: 6FNW) of this region as a model (*Deneka et al., 2018*).

## DCPIB inhibits LRRC8A through a cork-in-bottle mechanism

The extracellular cap of LRRC8A is formed by the TM1-TM2 and TM3-TM4 connections from each chain, which pack into a compact domain consisting of a three-stranded beta sheet (ES1-3) and short alpha helix EH1 stabilized by three intra-chain disulfide bonds (*Figure 1D,E*). The subunits associate to form a tube extending ~25 Å above the membrane surface with the inner surface of the tube formed by helix EH1. The N-terminal arginine residue of this helix (R103) projects in towards the conduction axis to form the highly electropositive selectivity filter (*Deneka et al., 2018*) of the anion-selective channel (*Figure 2*). The electropositive character in this region is also contributed by the helical dipole of EH1 projecting toward the conduction pathway.

A prominent density in the reconstruction, visible even in two-dimensional class averages, is a champagne cork-shaped feature found within and immediately above the R103 ring (*Figure 2A*). We attribute this bi-lobed density to the DCPIB inhibitor based on size, shape, chemical considerations, and comparison to apo-LRRC8A reconstructions (*Figure 2A* and below). Attempts to visualize a single binding pose for DCPIB by focused refinement, asymmetric refinement, or symmetry expansion were unsuccessful. Therefore, the density in our maps represents a six-fold average of positions adopted by DCPIB in different particles (*Figure 2B,C*). The density is well fit with the predominantly hydrophobic indane, butyl, and cyclopentyl constituents in the larger lobe extracellular to the R103 ring and the negatively charged butanoic acid group in the smaller lobe adjacent to and below the guanidinium groups of the R103 ring. In this way, the electronegative portion of DCPIB interacts favorably with the electropositive arginine side chains (*Figure 2C*). The connection between the two lobes corresponds to the ester linkage between the indane and butanoic acid. The weaker density for this region is presumably due to it adopting different conformations in different particles and the imposed six-fold symmetry.

To increase our confidence in the modeled DCPIB-binding site, we asked whether similar density features could be observed in apo-LRRC8A structures. We determined the structure of apo-LRRC8A in MSP2N2 nanodiscs and POPC lipids in a constricted conformation to 4.18 Å resolution (*Figure 1—*

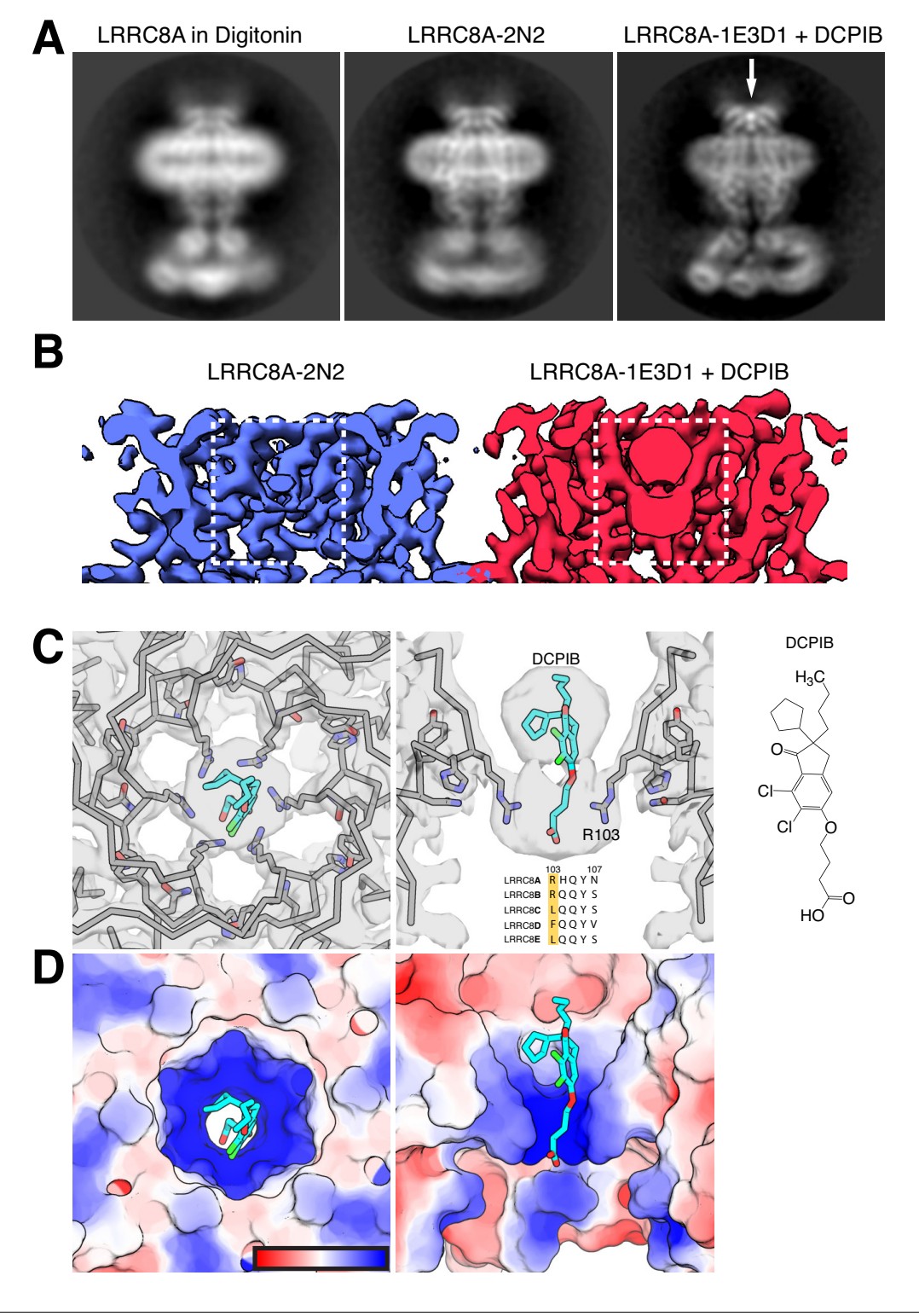

**Figure 2.** DCPIB inhibitor binding site. (**A**) Representative side-view two-dimensional class averages of LRRC8A (left) solubilized in digitonin, (middle) reconstituted in MSP2N2 lipid nanodiscs, or (right) MSP1E3D1 lipid nanodiscs and complexed with DCPIB. An arrow highlights the bi-lobed feature corresponding to DCPIB in the channel selectivity filter. (**B**) Cryo-EM density maps of the LRRC8A selectivity filter in (left) MSP2N2 nanodiscs or (right) MSP1E3D1 nanodiscs with DCPIB. The DCPIB binding region is highlighted in white boxes for each map. (**C**) View of the selectivity filter with bound DCPIB from (left) the extracellular solution (top view) and (middle) the

*Figure 2 continued on next page*

*Figure 2 continued*

membrane plane (side view). The atomic model is shown as ribbons and sticks within the cryo-EM density with the two front and two rear subunits removed in the side view for clarity. Nitrogens are colored blue, oxygens red, chlorines green, protein carbons gray, and DCPIB carbons teal. Alignment of the residues surrounding the selectivity filter for LRRC8 paralogs is shown below the drug density with numbering for LRRC8A above. (right) The chemical structure of DCPIB. (D) Views of the DCPIB-binding site as in (B), but with the atomic surface colored by electrostatic potential from electronegative red ($-5$ $k_bTe_c^{-1}$) to electropositive blue ($+5$ $k_bTe_c^{-1}$), with the color scale drawn on the left panel.

*figure supplement 8–10* and *Figure 2A,B*). Two-dimensional class averages and the final map of apo-LRRC8A lack the strong bi-lobed density along the conduction axis near the R103 ring

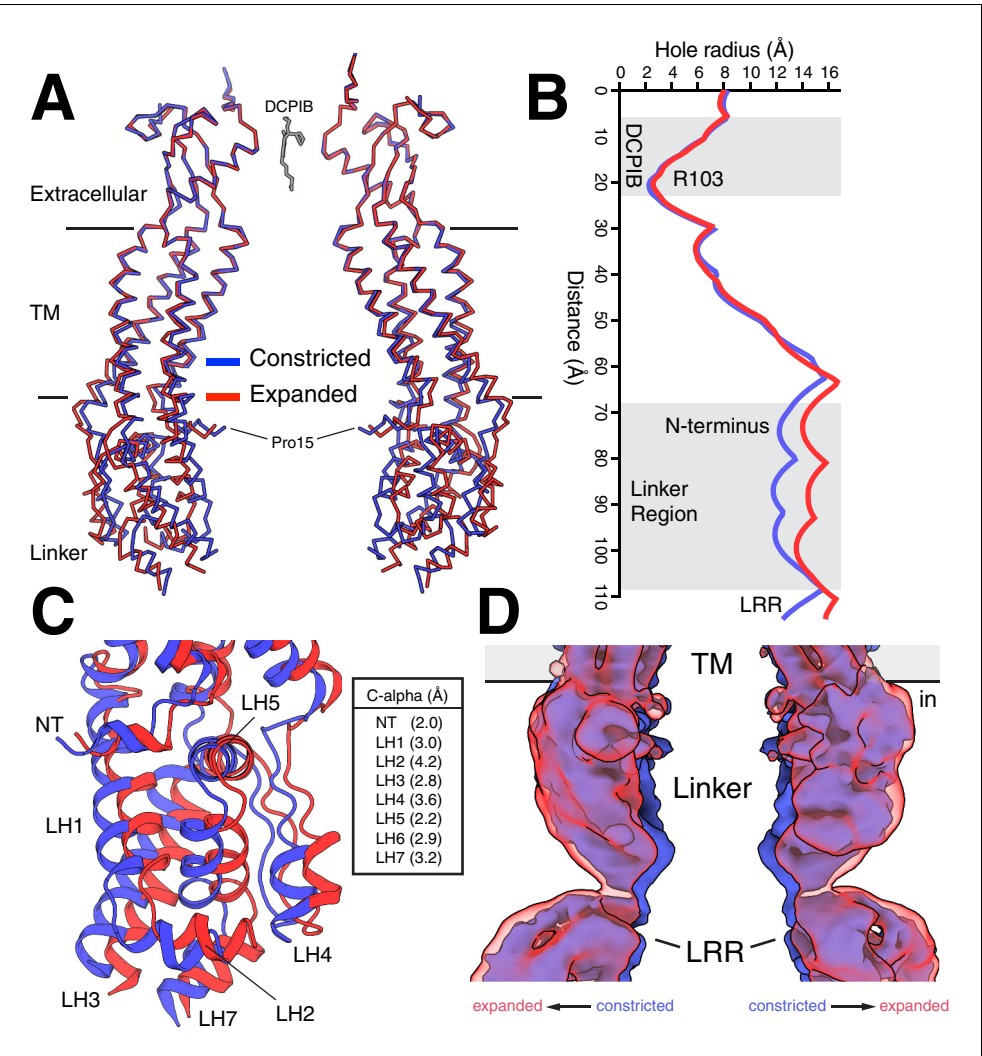

**Figure 3.** Constricted and expanded LRRC8A structures. (A) Overlay of constricted (blue) and expanded (red) structures of LRRC8A viewed from the membrane with two opposing subunits shown for each structure in ribbon representation. DCPIB is shown in grey sticks. Proline 15, the final modeled residue at the N-terminus, is labeled. (B) Comparison of the pore radius along the conduction axis colored as in (A). (C) Close-up view of the structure overlay at the linker region with models drawn as cartoons. Helices are labeled and distances between Cα positions at the following positions are indicated: NT, R18; LH1, K162; LH2, T170; LH3, K249; LH4, I356; LH5, S379; LH6, S387; LH7, E399. (D) Overlay of unmasked cryo-EM maps from constricted and expanded particles showing correlated movement of the linker region and membrane-proximal LRR region. Density within 5 Å of two opposing chains is shown.

**Table 1.** Cryo-EM data collection, processing, refinement, and modeling data for LRRC8A-DCPIB in MSP1E3D1 nanodiscs and LRRC8A in MSP2N2 nanodiscs.

| | LRRC8A-MSP1E3D1 + DCPIB | | LRRC8A-MSP2N2 |
|---|---|---|---|
| **Data collection** | **Sloan-Kettering** | **Nysbc** | **Nysbc** |
| Movie # | 2482 | 2110 | 1786 |
| Magnification | 22,500x | 22,500x | 22,500x |
| Voltage (kV) | 300 | 300 | 300 |
| Electron exposure ($e^-/\text{Å}^2$) | 60.8 | 55.6 | 70.3 |
| Defocus range (μm) | −1.2 ~ −2.5 | −1.2 ~ −2.5 | −1.2 ~ −2.5 |
| Super resolution pixel size (Å) | 0.544 | 0.536 | 0.536 |
| Fourier cropped pixel size (Å) | 1.088 | 1.088 | 1.088 (1.149 *Figure 4*) |
| **Processing** | **Class 1 (constricted)** | **Class 2 (expanded)** | **Class 1 (constricted)** |
| Symmetry imposed | C6 | C6 | C6 |
| Initial particle images (no.) | 752,736 | 752,736 | 252, 655 |
| Final particle images (no.) | 25,153 | 35,435 | 11,507 |
| Map resolution (umasked, Å) /FSC threshold | 3.39/0.143 | 3.63/0.143 | 4.28/0.143 |
| Map resolution (masked, Å) /FSC threshold | 3.21/0.143 | 3.32/0.143 | 4.18/0.143 |
| **Refinement** | | | |
| Model resolution (Å) | 3.52/3.32 | 3.81/3.47 | 4.4/3.8 |
| FSC threshold | 0.50/0.143 | 0.50/0.143 | 0.50/0.143 |
| Map-sharpening *B*factor ($\text{Å}^2$) | −44.538 | −134.6 | −82.8 |
| Ligands | 19 | 19 | 0 |
| **Mean *B* factors ($\text{Å}^2$)** | | | |
| Protein | 87.73 | 52.17 | 152.09 |
| Ligand | 65.05 | 27.6 | - |
| **R.m.s. deviations** | | | |
| Bond lengths (Å) | 0.007 | 0.005 | 0.004 |
| Bond angles (°) | 0.775 | 0.823 | 0.773 |
| **Validation** | | | |
| MolProbity score | 1.74 | 1.94 | 1.31 |
| Clashscore | 3.34 | 4.12 | 2.16 |
| Poor rotamers (%) | 3.09 | 3.17 | 0.69 |
| EMRinger score | 2.7 | 2.2 | 0.7 |
| **Ramachandran plot** | | | |
| Favored (%) | 96.42 | 94.67 | 95.44 |
| Allowed (%) | 3.58 | 5.33 | 4.56 |
| Disallowed (%) | 0 | 0 | 0.69 |

attributed to DCPIB in drug-bound LRRC8A (*Figure 2A,B*). This density is similarly absent from four separate reconstructions of apo-LRRC8A in detergent: a low-resolution reconstruction presented here (*Figure 2A*) and three independently determined to higher resolution (*Deneka et al., 2018*; *Kasuya et al., 2018*; *Kefauver et al., 2018*).

The structure of DCPIB bound to LRRC8A provides a simple explanation for the mechanism of drug block: DCPIB acts like a cork to plug the mouth of the channel. The R103 ring electrostatically interacts with the DCPIB carboxylic acid while the bulky hydrophobic end of the drug is too large to pass. Ion conduction is thus prevented by an obstructed selectivity filter. DCPIB universally blocks

**Table 2.** Cryo-EM data collection information for the digitonin datasets used in *Figure 2A* and *Figure 4*.

| Data collection | Digitonin 70 mM KCl | Digitonin 150 mM KCl | Digitonin 600 mM KCl |
|---|---|---|---|
| Movie # | 2550 | 1445 | 1464 |
| Magnification | 22,500x | 22,500x | 22,500x |
| Voltage (kV) | 300 | 300 | 300 |
| Electron exposure (e⁻/Å²) | 60.8 | 55.6 | 55.6 |
| Defocus range (µm) | −1.2 ~ −2.5 | −1.2 ~ −2.5 | −1.2 ~ −2.5 |
| Super resolution pixel size (Å) | 0.536 | 0.544 | 0.544 |
| Fourier cropped pixel size (Å) | 1.088 | 1.088 | 1.088 |

LRRC8 currents even though amino acids around the inhibitor binding site in LRRC8A (including R103) are not strictly conserved in LRRC8A paralogs (*Figure 2B*, right). However, since all functional LRRC8 channels contain LRRC8A subunits, similar drug-LRRC8A interactions could account for block of heteromeric channels. Consistent with this model of channel inhibition by anionic small molecules, an R103F mutation in LRRC8A that renders the selectivity filter ring apolar in heteromeric LRRC8AC channels eliminates extracellular block by 1 mM ATP (*Kefauver et al., 2018*). Electrostatic and steric complementarity around the R103 ring are therefore important determinants of LRRC8 block by anionic inhibitors.

## Constricted and expanded structures provide a mechanism to relay conformational changes between an ionic strength sensor and channel gate

Focused classification on the linker region and lower portion of the TMs resolved a constricted and an expanded state of LRRC8A (*Figure 3A* and *Figure 1—figure supplement 4*). The structure of the extracellular cap, DCPIB-bound selectivity filter, and extracellular halves of the TM region are essentially indistinguishable between the two structures (mean Cα r.m.s.d. = 0.42 Å). Alignment of the two structures by the extracellular cap shows that the differences are confined to the cytoplasmic halves of the TMs, linker regions, and LRRs. The differences can be approximated as a rigid body displacement of the lower portion of the channel about hinges approximately half way through each transmembrane helix. This generates displacements of ~2–4 Å in the cytoplasmic ends of the transmembrane helices, the linkers, and at the cytoplasmic connection between the linkers and LRRs (*Figure 3B–D*). As a consequence of these conformational changes, the channel cavity is markedly larger in the expanded state: it dilates approximately 4 Å including at the first resolved N-terminal residue (Pro15) (*Figure 3B* and *Videos 1* and *2*).

Importantly, the presence of the two conformations is not a consequence of drug-binding or imposed symmetry in reconstructions. The two states are comparably populated in apo-LRRC8A and DCPIB-bound nanodisc datasets classified with three different approaches (*Figure 4—figure supplements 1,2*) and structures of constricted classes from apo-LRRC8A and DCPIB-bound LRRC8A in nanodiscs are not significantly different in the extracellular, transmembrane, and linker regions (mean Cα r.m.s.d. = 0.32 Å, *Figure 1—figure supplement 9D*).

Might the differences between constricted and expanded states be related to gating conformational changes? Gating in LRRC8 and related connexin and innexin channels involves the N-terminus prior to the beginning of TM1 (*Kefauver et al., 2018*; *Oshima, 2014*; *Zhou et al., 2018*), whereas sensing of internal ionic strength is thought to occur in the cytoplasmic LRRs or connection between linker helices 2 and 3 (*Syeda et al., 2016*). Notably, the conformational changes we observe between constricted and expanded states couple these presumed sensing and gating elements. The extreme N-terminus of LRRC8A (amino acids 1–14) is not resolved in our structures (or in detergent-solubilized LRRC8A structures [*Deneka et al., 2018*; *Kasuya et al., 2018*; *Kefauver et al., 2018*]) presumably due to flexibility that results in heterogeneity between particles. However, amino acids 15–21 are visible and form a helical extension of TM1 that projects toward the center of the channel (*Figure 3A and B*). This region is coupled through the linkers to the LRRs via an elbow formed by the N-terminal helix, TM1, and LH5 (*Figure 3C*). The conformational changes we observe show how expansion of cytoplasmic ionic-strength sensing regions can be coupled to expansion of N-terminal

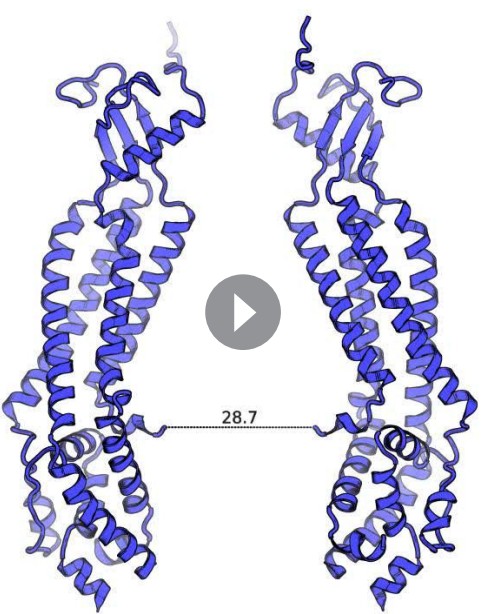

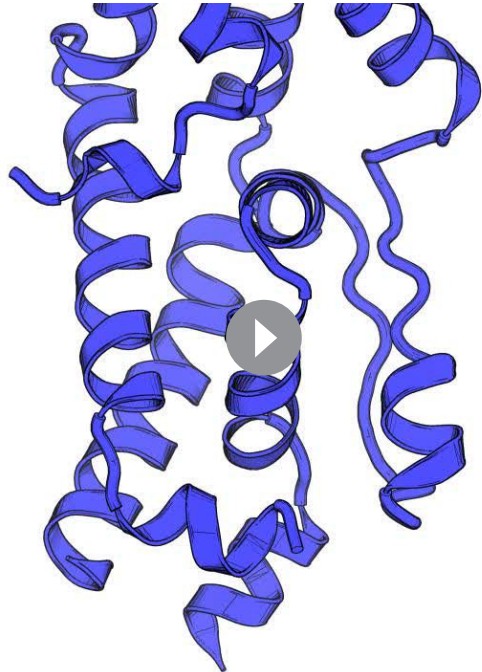

**Video 1.** Motion of two opposing chains (as in *Figure 3A*) as a cartoon-representation morph between constricted and expanded states. Measurement in Å between the C-alpha of Pro15 is included.

https://elifesciences.org/articles/42636#video1

**Video 2.** Linker region (as in *Figure 3C*) motion as a cartoon-representation morph between constricted and expanded states.

https://elifesciences.org/articles/42636#video2

gating regions and the channel pore. Whether these changes are alone sufficient to gate the channel or rather represent intermediates along a pathway of larger gating conformational changes depends in part upon the position of extreme N-terminus.

## Channel symmetry and heterogeneity of LRRs in nanodiscs

Recent structures of LRRC8A in digitonin showed an unexpected three-fold symmetric trimer of dimers arrangement with pairs of LRRC8A subunits forming an asymmetric unit of the channel (*Deneka et al., 2018*; *Kasuya et al., 2018*; *Kefauver et al., 2018*). Strikingly, LRRC8A in lipid nanodiscs does not display this arrangement. Instead, we observe six-fold symmetric channels with conformationally heterogeneous LRRs.

Structural heterogeneity in LRRs of particles in lipid nanodiscs is readily apparent in 2D class averages (*Figure 4A*, *Videos 3* and *4*). Side views of the particles constituting both constricted and expanded classes display a range of LRR positions: LRRs are closely packed underneath the linker region in some classes and splayed laterally away from the conduction axis in others. As expected from the two-dimensional classes, three-dimensional reconstructions with or without symmetry enforced, or using masking strategies to isolate the LRRs, generated low-resolution features for this region. We conclude that LRRs of LRRC8A-DCPIB in lipid nanodiscs can access a large conformational space. Interestingly, comparison of two-dimensional class averages of apo-LRRC8A and DCPIB-bound LRRC8A in nanodiscs shows an apparent increase in the conformational space sampled by the LRRs in the presence of DCPIB (*Figure 4A* and *Figure 4—figure supplement 3*). Still, the average position of membrane proximal region of the LRRs adopts a more constricted or expanded position in the corresponding structure class, about which there appears to be similar heterogeneity (*Figure 3D* and *Figure 4A*).

We used four approaches to assess the symmetry of LRRC8A in lipid nanodiscs. First, we subjected LRRC8A-nanodisc particle stacks (generated without imposing symmetry) to classification without symmetry (C1), with three-fold (C3), or with six-fold (C6) symmetry imposed (*Figure 4B*, *Figure 4—figure supplements 1–3*). Reconstructions without imposed symmetry have six-fold

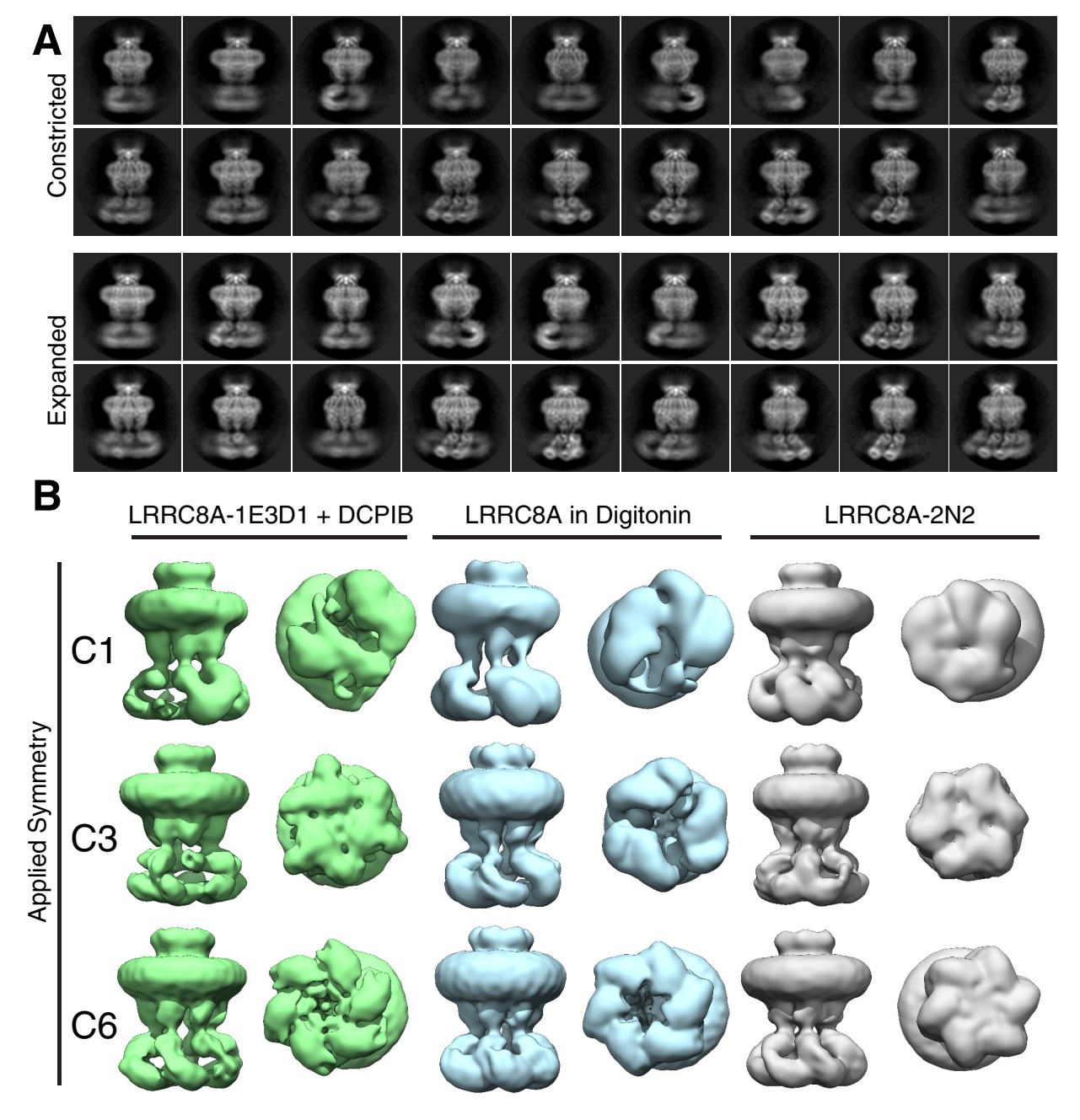

**Figure 4.** LRR position and channel symmetry differences in lipid and detergent environments. (**A**) Side-views of two-dimensional class averages from the (top) constricted and (bottom) expanded particle classes illustrating variation in LRR position. Also see *Videos 3* and *4*. (**B**) Symmetry comparison of LRRC8A in (left) MSP1E3D1 + DCPIB, (middle) digitonin, or (right) MSP2N2 (displayed at a 0.015 threshold). Selected classes from three-dimensional classing jobs with the indicated symmetry are shown from the side (membrane) or bottom (cytoplasm).

The online version of this article includes the following figure supplement(s) for figure 4:

**Figure supplement 1.** Full three-dimensional classification output for symmetry testing.

**Figure supplement 2.** Classification of constricted and expanded states.

**Figure supplement 3.** Initial processing for symmetry testing.

symmetric features, including in the linker region where individual subunits are oriented and spaced uniformly, that are recapitulated with improved definition and resolution when three- and six-fold symmetry is imposed. Second, the final LRRC8A-nanodisc particles used for high-resolution

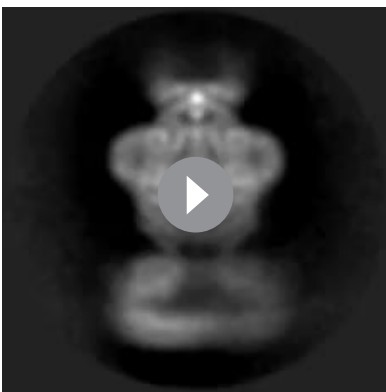

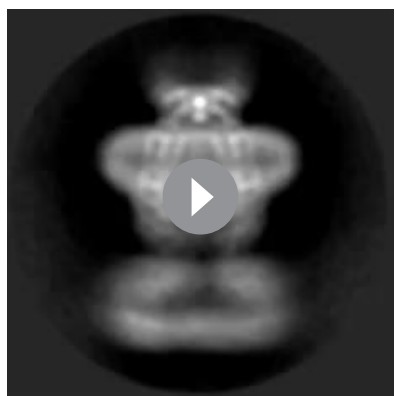

**Video 3.** Constricted state side-views of LRRC8A-DCPIB in MSP1E3D1 nanodiscs illustrating heterogenous LRR positions.
https://elifesciences.org/articles/42636#video3

**Video 4.** Expanded state side-views of LRRC8A-DCPIB in MSP1E3D1 nanodiscs illustrating heterogenous LRR positions.
https://elifesciences.org/articles/42636#video4

reconstructions (generated with C6-symmetry imposed and LRRs masked) were refined with C1, C3, or C6 symmetry. Again, C1 reconstructions display features with six-fold symmetry and imposing C3 and C6 symmetry improves these map features. Third, we attempted to identify a class of C3 symmetric particles in nanodiscs using references made from reported C3-symmetric structures of LRRC8A in digitonin. This approach failed to generate high-resolution reconstructions with or without masking of the LRRs. Finally, we performed symmetry expansion of refined particles and subjected them to asymmetric classification without alignment. This approach results in similar approximate six-fold symmetric reconstructions. We therefore conclude that LRRC8A-DCPIB in nanodiscs is six-fold symmetric outside of the heterogeneous LRRs.

To determine whether differences between nanodisc and digitonin structures are related to different hydrophobic environments, drug binding, or other factors, we performed analogous analyses of apo-LRRC8A in MSP2N2 nanodiscs and apo-LRRC8A in digitonin (*Figure 4B* and *Figure 4—figure supplements 1–3*). Drug binding had no effect on symmetry or overall channel structure apart from increasing the conformational space sampled by the LRRs (*Figure 4* and *Figure 4—figure supplement 3*). In contrast, apo-LRRC8A particles in digitonin display three-fold symmetry with compact LRRs (i.e. a trimer-of-dimers arrangement), consistent with published structures (*Deneka et al., 2018*; *Kasuya et al., 2018*; *Kefauver et al., 2018*). Thus, the presence of DCPIB, choice of scaffold protein, or expression host are unlikely to generate structural and symmetry differences in LRRC8A. Instead, we conclude that the hydrophobic environment surrounding LRRC8A is a key determinant of channel structure: digitonin micelles promote three-fold channel symmetry and compact LRRs while lipid bilayers promote six-fold symmetric channels with asymmetric and conformationally heterogeneous LRRs.

How might the differences between digitonin and lipid bilayers influence the structure of LRRC8A channels? A conspicuous feature of LRRC8A is the presence of gaps between subunits that create lipid-facing crevasses in the channel surface. In lipid nanodiscs, a large lower gap and a smaller upper gap are separated by a constriction made by amino acids Leu131 and Phe324. The upper gap is filled with three well-defined tubular features that are modeled as partial POPC lipid hydrocarbon acyl chains (*Figure 5*). The presence of these ordered acyl chains seals the upper gap in the channel surface and may act as a 'glue' that connects adjacent subunits and stabilizes the upper transmembrane domain against movements in channel linkers and LRRs. In the C3-symmetric LRRC8A structures in digitonin (*Kasuya et al., 2018*; *Kefauver et al., 2018*), there are two distinct subunit interfaces – one with a narrow separation and one with a wider separation between neighboring chains (*Figure 6* and *Videos 5* and *6*). This results in a striking rearrangement of the lipid-binding site. In nanodiscs, the upper gap tapers to a tunnel ideally sized to surround the central bound acyl chain (5–6 Å in diameter). In detergent, in contrast, this gap becomes either too large to optimally accommodate an acyl chain (6.9–7.3 Å diameter in the wide interface) or too small for one to fit (4.1 Å diameter in the narrow interface). Consistently, density for lipids is not observed in LRRC8A-

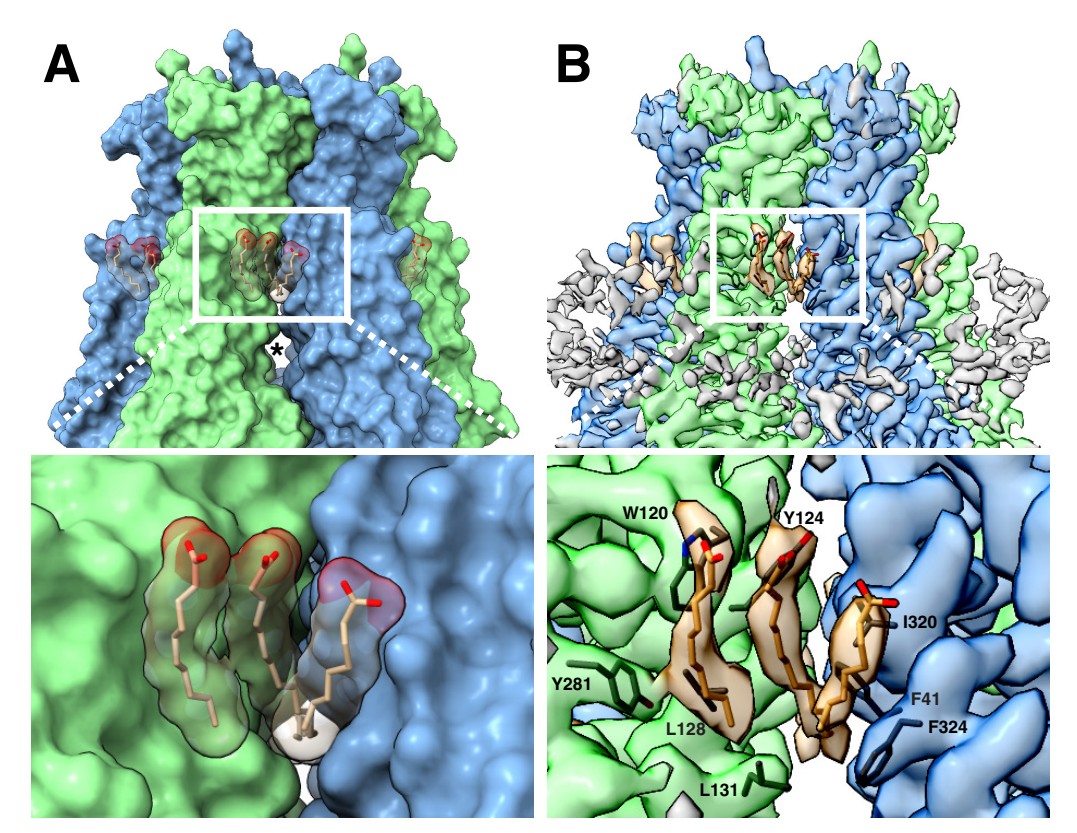

**Figure 5.** LRRC8A-lipid interactions. (**A**, above) Surface representation of the constricted LRRC8A class viewed from the membrane with docked POPC lipid chains depicted in stick (tan) and space-filling (transparent) representations. The upper gap between subunits is filled by lipid density and an asterisk marks the larger lower gap. (Below) a zoomed-in view on the upper gap and bound lipid. (**B**, above) The corresponding cryo-EM density viewed as in (**A**), with lipid and the nearby hydrophobic amino-acid side chains shown in stick form. (Below), a zoomed-in view with the amino acids of the hydrophobic pocket labeled.

digitonin reconstructions. In one map, a feature consistent with a digitonin molecule is wedged between subunits near the wide-interface upper gap instead (*Figure 6C*) (*Kefauver et al., 2018*). This suggests an explanation for the differences between LRRC8A structures: when lipid surrounding the channel is removed, the channel adapts from six evenly spaced lipid-filled gaps to three smaller gaps incompatible with lipid binding and three larger gaps perhaps filled with digitonin. These changes at subunit interfaces in the transmembrane region are propagated through the structure to influence overall channel structure and symmetry.

## Discussion

The nanodisc-embedded LRRC8A-DCPIB structures presented here provide the first insights into drug block of LRRC8 channels or related connexin and innexin channels. DCPIB binding in the channel selectivity filter is largely dictated by size and electrostatic complementarity of the small molecule and channel. The DCPIB-binding site appears to be relatively rigid; there are no detectable conformational changes in the extracellular cap, transmembrane region, or linkers between drug-free and DCPIB-bound constricted state structures and no detectable changes in cap structure between any structure in nanodisc or detergent (although we can not rule out the possibility of changes in the cap of expanded state apo-LRRC8A in nanodiscs in the absence of a high-resolution reconstruction). We note that in nanodiscs, drug binding does appear to increase conformational heterogeneity of LRRs in two-dimensional class averages, suggesting the possibility of allosteric communication between the cytoplasmic and extracellular regions. The functional ramifications of this increase in LRR flexibility, if any, remain to be determined. The small LRRC8A-drug interface

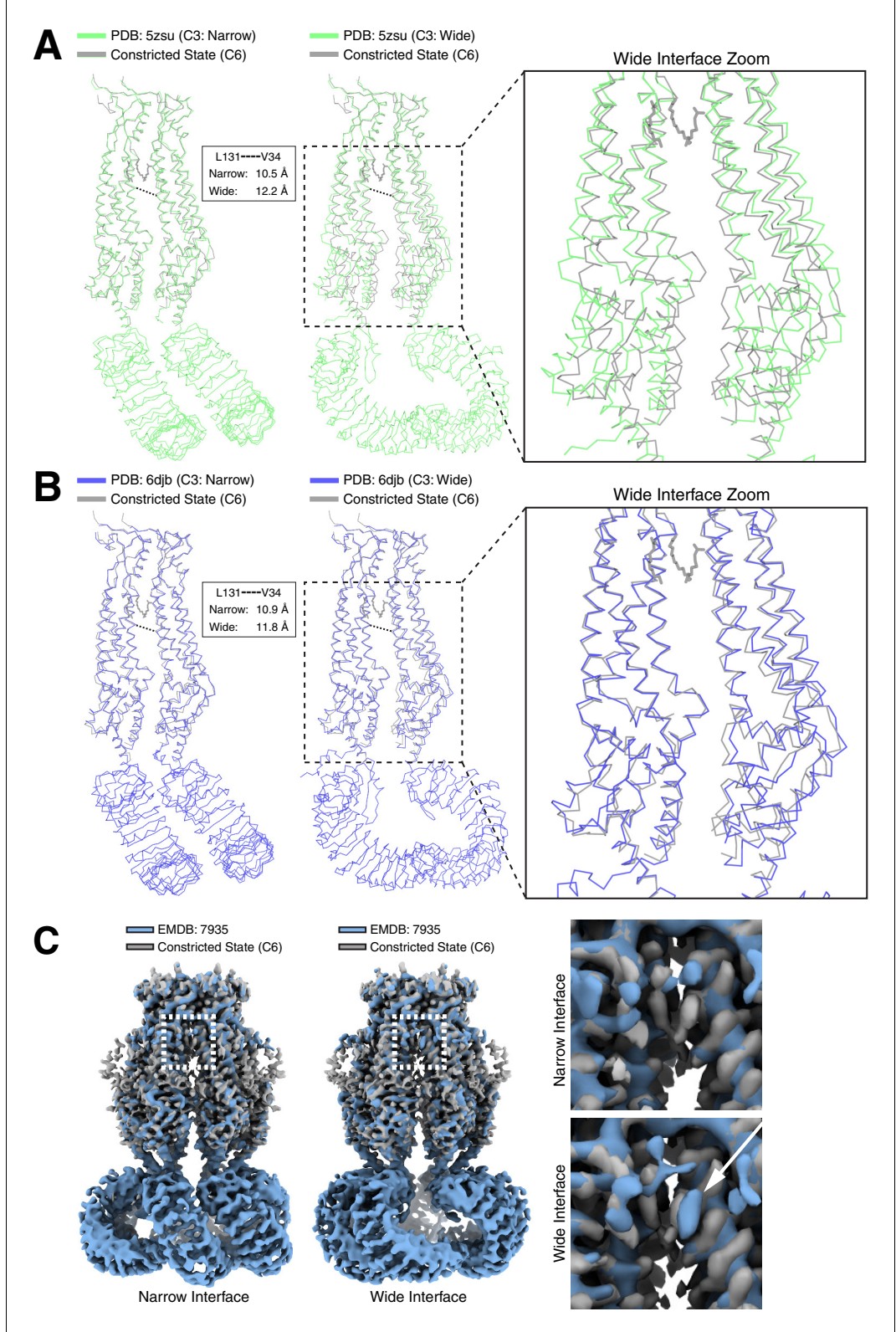

**Figure 6.** Differences in LRRC8A structures solved in lipid bilayers and detergent micelles. Overlays of extracellular domain-aligned models of the constricted state structure determined in lipid nanodiscs (gray) and the (left) narrow and (middle) wide subunit interfaces from structures determined in digitonin: (A, PDB: 5zsu, green [*Kasuya et al., 2018*]), (B, PDB: 6djb, blue [*Kefauver et al., 2018*]). Boxed and zoomed in regions (right) illustrate the expansion in subunits of the wide interfaces extending to the LRR region. Distances between V34 and L131 Cαs in detergent-solubilized structures are
*Figure 6 continued on next page*

*Figure 6 continued*

indicated. (C) Aligned density between the six-fold symmetric constricted state in nanodiscs (gray) and LRRC8A in digitonin (Blue, EMDB: 7935, [*Kefauver et al., 2018*]). (left) View from the membrane of the narrow C3 subunit interface. (middle) View from the membrane of the wide C3 subunit interface. Boxed region and zoomed-in panels (right) show the upper gap filled with lipid density in nanodiscs or an extra density consistent with digitonin in the wide interface marked by a white arrow.

likely primarily accounts for the relatively modest (low micromolar) affinity of DCPIB for LRRC8 channels (*Decher et al., 2001*). This is also consistent with the broad target profile of DCPIB; in addition to inhibiting LRRC8 homomeric and heteromeric channels, it inhibits connexin hemichannels (*Bowens et al., 2013*). The structures reported here provide a platform for future structure-based design of inhibitors with higher affinity or more restricted target range, perhaps by increasing the drug-channel interface size and complementarity in the extracellular cap. Improved pharmacology would likely help to both refine proposed and define new physiological roles for LRRC8 channels.

The correlated changes in the N-terminal region, linker, and LRRs in the constricted and expanded states suggest a mechanism to transmit information from the ionic strength sensor to the channel gate. To our knowledge, there is currently no experimental evidence that supports a model in which the expanded state is more open than the constricted state (or vice versa). Still, it is tempting to speculate based on structural considerations that the expanded state represents an intermediate on the path to a fully conductive channel. The 4 Å increase in channel cavity diameter at the position of the unresolved N-terminus represents a significant expansion, comparable in size to the selectivity filter opening and approaching the diameter of a solvated chloride ion. Still, we cannot determine whether these changes are sufficient to fully open the channel since the N-terminal 14 amino acids that project into the channel cavity are unresolved. Since the particles are vitrified in an ionic strength that would correspond to closed or inactivated channels (150 mM KCl), additional conformational changes might be expected to fully open the channel. One possibility is that a dynamic N-terminal region creates an entropic barrier to ion passage in nonconductive channels and gating by low ionic strength involves conformational changes that order the N-terminus to create a favorable environment for anion conduction.

Intriguingly, both the symmetry changes and conformational flexibility of the LRRs are reminiscent of the unrelated bacterial CorA channel (*Deneka et al., 2018*; *Kefauver et al., 2018*; *Matthies et al., 2016*). In CorA, loss of intracellular $Mg^{2+}$ binding allows intracellular domains to move outward and adopt an asymmetric conformation that promotes conduction. It is not known how LRRC8A senses reduced internal ionic strength. It may be that specifically bound ions promote the closed state in physiological ionic strength. Alternatively, polar or ionic protein interactions could be stabilized when ionic strength is sufficiently high to provide charge screening. In either case, reduced ionic strength may result in loss of interactions between cytoplasmic LRRs and/or linker helix 2–3 connections that otherwise promote a compact structure and constricted pore. This could allow an expansion in cytoplasmic regions that is relayed through the linkers to dilate and open the channel. Future studies that define conductive LRRC8 structures and correlate channel motions with functional states will help to further elucidate ionic-strength sensing and gating mechanisms of this dynamic and prolific vertebrate channel family.

## Data availability

Final maps of LRRC8A-DCPIB in MSP1E3D1 nanodiscs have been deposited to the Electron Microscopy Data Bank under accession codes EMDB-0562 (masked constricted state) and EMDB-0563 (masked expanded state). Atomic coordinates have been deposited in the PDB under IDs 6NZW (constricted state) and 6NZZ (expanded state). The original micrograph movies have been deposited to EMPIAR under accession codes EMPIAR-10258 and EMPIAR-10259. The map of apo-LRRC8A in MSP2N2 nanodiscs in a constricted state has been deposited with EMDB accession code EMDB-0564 and coordinates deposited in the PDB with ID 6O00.

## Materials and methods

### Key resources table

| Reagent type (species) or resource | Designation | Source or reference | Identifiers | Additional information |
|---|---|---|---|---|
| Gene (*Mus musculus*) | LRRC8A | Gen9 synthesis | Uniprot: Q80WG5 | Codon-optimized for *Spodoptera frugiperda* |
| Cell Line (*Spodoptera frugiperda*) | Sf9 | Expression Systems | Catalog Number: 94–001F | |
| Peptide, recombinant protein | MSP1E3D1 | Prepared as described in doi: 10.1016/S0076-6879(09)64011–8 | | His-tag cleaved |
| Peptide, recombinant protein | MSP2N2 | Prepared as described in doi: 10.1016/S0076-6879(09)64011–8 | | His-tag cleaved |
| Chemical compound, drug | DDM | Anatrace | Part Number: D310S | |
| Chemical compound, drug | CHS | Anatrace | Part Number: CH210 | |
| Chemical compound, drug | Digitonin | EMD Chemicals | CAS 11024-24-1 | |
| Chemical compound, drug | 16:0-18:1 PC (POPC) lipid | Avanti Polar Lipids | SKU: 850457C | |
| Chemical compound, drug | DCPIB | Tocris | CAS Number: 82749-70-0, Catalog Number: 1540 | |
| Software, algorithm | RELION | DOI: 10.7554/eLife.42166 | Relion 3.0 | |
| Software, algorithm | Gctf | DOI: 10.1016/j.jsb.2015.11.003 | Gctf v1.06 | |
| Software, algorithm | UCSF Chimera | UCSF | RRID:SCR_004097 | http://plato.cgl.ucsf.edu/chimera/ |
| Software, algorithm | COOT | | RRID:SCR_014222 | http://www2.mrc-lmb.cam.ac.uk/personal/pemsley/coot/ |
| Software, algorithm | Phenix | | RRID:SCR_014224 | https://www.phenix-online.org/ |
| Software, algorithm | PyMOL | PyMOL Molecular Graphics System, Schrodinger LLC | RRID:SCR_000305 | https://www.pymol.org/ |

## Protein expression

The coding sequence for LRRC8A from *Mus musculus* was codon optimized for *Spodoptera frugiperda* and synthesized (Gen9, Cambridge, MA). The sequence was then cloned into a custom vector based on the pACEBAC1 backbone (MultiBac; Geneva Biotech, Geneva, Switzerland) with an added C-terminal PreScission protease (PPX) cleavage site, linker sequence, superfolder GFP (sfGFP) and 7xHis tag, generating a construct for expression of mmLRRC8A-SNS-LEVLFQGP-SRGGSGAAAGSG SGS-sfGFP-GSS-7xHis. MultiBac cells were used to generate a Bacmid according to manufacturer's instructions. *Spodoptera frugiperda* (Sf9) cells were cultured in ESF 921 medium (Expression Systems, Davis, CA) and P1 virus was generated from cells transfected with Cellfectin II reagent (Life Technologies, Carlsbad, CA) according to manufacturer's instructions. P2 virus was then generated by infecting cells at 2 million cells/mL with P1 virus at an MOI ~ 0.1, with infection monitored by fluorescence of sfGFP-tagged protein and harvested at 72 hr. P3 virus was generated in a similar manner to expand the viral stock. The P3 viral stock was then used to infect 1 L of Sf9 cells at 4 million cells/

mL at an MOI ~ 2–5. At 72 hr, infected cells containing expressed LRRC8A-sfGFP protein were harvested by centrifugation at 2500 x g and frozen at −80°C.

## Protein purification

Cells from 1 L of culture (~15–20 mL of cell pellet) were thawed in 100 mL of Lysis Buffer containing (in mM) 50 HEPES, 150 KCl, 1 EDTA pH 7.4. Protease inhibitors (Final Concentrations: E64 (1 μM), Pepstatin A (1 μg/mL), Soy Trypsin Inhibitor (10 μg/mL), Benzimidine (1 mM), Aprotinin (1 μg/mL), Leupeptin (1 μg/mL), and PMSF (1 mM)) were added to the lysis buffer immediately before use. Benzonase (4 μl) was added after cell thaw. Cells were then lysed by sonication and centrifuged at 150,000 x g for 45 min. The supernatant was discarded and residual nucleic acid was removed from the top of the membrane pellet using DPBS. Membrane pellets were scooped into a dounce homogenizer containing Extraction Buffer (50 mM HEPES, 150 mM KCl, 1 mM EDTA, 1% n-Dodecyl-β-D-Maltopyranoside (DDM, Anatrace, Maumee, OH), 0.2% Cholesterol Hemisuccinate Tris Salt (CHS, Anatrace) final pH 7.4). A 10%/2% solution of DDM/CHS was dissolved and clarified by bath sonication in 200 mM HEPES pH eight prior to addition to buffer to the indicated final concentration. Membrane pellets were then homogenized in Extraction Buffer and this mixture (150 mL final volume) was gently stirred at 4°C for 3 hr. The extraction mixture was centrifuged at 33,000 x g for 45 min and the supernatant, containing solubilized membrane protein, was bound to 4 mL of sepharose resin coupled to anti-GFP nanobody for 1 hr at 4°C. The resin was then collected in a column and washed with 10 mL of Buffer 1 (20 mM HEPES, 150 mM KCl, 1 mM EDTA, 0.025% DDM, pH 7.4), 40 mL of Buffer 2 (20 mM HEPES, 500 mM KCl, 1 mM EDTA, 0.025% DDM, pH 7.4), and 10 mL of Buffer 1. The resin was then resuspended in 6 mL of Buffer 1 with 0.5 mg of PPX and rocked gently in the capped column for 2 hr. Cleaved LRRC8A protein was then eluted with an additional 8 mL of Wash Buffer, spin concentrated to ~500 μl with Amicon Ultra spin concentrator 100 kDa cutoff (Millipore), and then loaded onto a Superose 6 Increase column (GE Healthcare, Chicago, IL) on an NGC

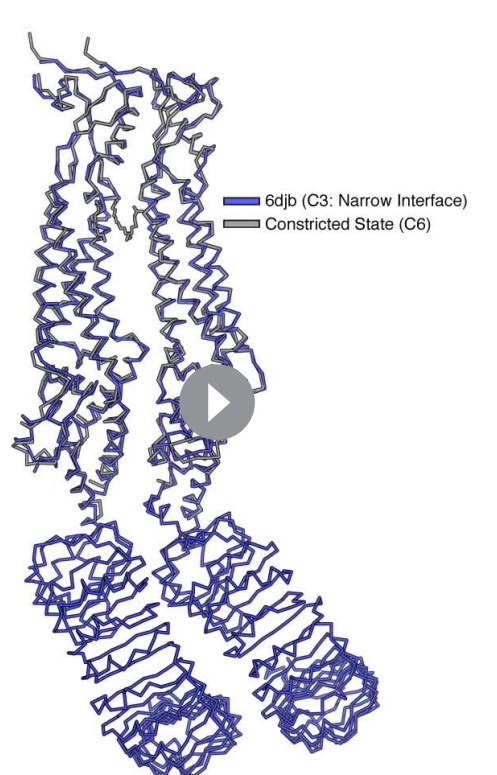

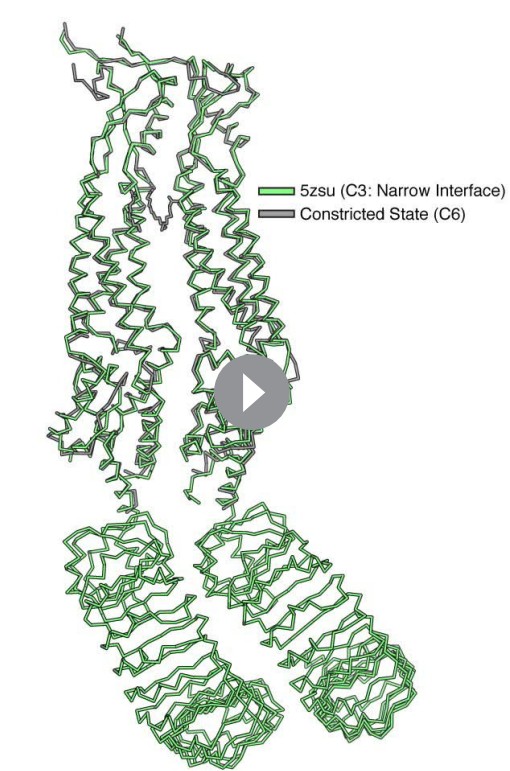

**Video 5.** Overlay of the constricted state and narrow and wide interfaces of PDB: 6djb.
https://elifesciences.org/articles/42636#video5

**Video 6.** Overlay of the constricted state and narrow and wide interfaces of PDB: 5zsu.
https://elifesciences.org/articles/42636#video6

system (Bio-Rad, Hercules, CA) equilibrated in buffer 1. Peak fractions containing LRRC8A channel were then collected and spin concentrated.

Purification in digitonin was performed analogously with the following modifications. Tris buffer was used instead of HEPES buffer during purification. Washing and cleavage steps were carried out in 0.1% DDM and 0.02% CHS and cleavage was performed overnight. Digitonin (EMD Chemicals Inc, San Diego, CA) Buffer containing 20 mM Tris, (70, 150, or 600) mM KCl, 1 mM EDTA, 0.05% Digitonin (final pH 8) was prepared by dissolving digitonin in buffer at room temperature, cooling buffer to 4°C, and 0.2 μm filtering the buffer to remove insoluble material. Protein was exchanged into digitonin buffer by gel filtration in a Superose 6 Increase column. Fractions containing LRRC8A channel were pooled and spin concentrated.

## Electrophysiology

For proteoliposome patching experiments, we incorporated protein into lipid and generated proteoliposome blisters for patch recordings using dehydration and rehydration as described previously (*Brohawn et al., 2014*; *Del Mármol et al., 2018*) with the following modifications. LRRC8A was first purified into Column Buffer with DDM/CHS at 0.025%/0.005%. Protein was then exchanged into lipid with the addition of Biobeads SM2 and an overnight incubation. Dried proteoliposomes in a dish were rehydrated overnight in a buffer containing 10 mM HEPES, 70 mM KCl, pH 7.4. The next day, the dish was filled with a bath solution containing 10 mM HEPES, 20 mM MgCl$_2$, 30 mM KCl pH 7.4. The pipette solution was 10 mM HEPES, 70 mM KCl, pH 7.4. All experiments were conducted at room temperature.

## Nanodisc formation

Freshly purified LRRC8A from gel filtration in Buffer one was reconstituted into MSP1E3D1 nanodiscs with POPC lipid (Avanti, Alabaster, Alabama) at a final molar ratio of 1:2.5:250 (Monomer Ratio: LRRC8A, MSP1E3D1, POPC). First, solubilized lipid in Column Buffer (20 mM HEPES, 150 mM KCl, 1 mM EDTA pH 7.4) was mixed with additional DDM detergent, Column Buffer, and LRRC8A. This solution was mixed at 4°C for 30 min before addition of purified MSP1E3D1. This addition brought the final concentrations to approximately 10 μM LRRC8A, 25 μM MSP1E3D1, 2.5 mM POPC, and 4 mM DDM in Column Buffer. The solution with MSP1E3D1 was mixed at 4°C for 30 min before addition of 160 mg of Biobeads SM2 (Bio-Rad). Biobeads (washed into methanol, water, and then Column Buffer) were weighed with liquid removed by P1000 tip (Damp weight). This mix was incubated at 4°C for 30 min before addition of another 160 mg of Biobeads (final 320 mg of Biobeads per mL). This final mixture was then mixed at 4°C overnight (~12 hr). Supernatant was cleared of beads by letting large beads settle and by 0.2 μm filtering. Sample was spun for 5 min at 21,000 x g before loading onto a Superose six column in Column Buffer. Peak fractions corresponding to LRRC8A in MSP1E3D1 were collected, 100 kDa cutoff spin concentrated, and then re-run on the Superose 6. The fractions corresponding to the center of the peak were then pooled and concentrated prior to grid preparation. For MSP2N2 nanodiscs, reconstitution was carried out with a molar ratio of 1:2:300 (Monomer Ratio: LRRC8A, MSP2N2, POPC (Avanti)). The final 1 mL reaction mix contained 5 μM LRRC8A monomer, 10 μM MSP2N2, 1.5 mM POPC,~2.6 mM DDM, and 200 mg of Biobeads in Column Buffer. Column purification was performed similarly to the MSP1E3D1 preparation. Nanodisc proteins were prepared as described in *Ritchie et al. (2009)* and His-tags were cleaved with TEV protease.

## Grid preparation

For the MSP1E3D1 nanodisc samples, 100 μM of DCPIB (Tocris, Bristol, UK) was added to sample to give a final concentration of 0.8 mg/mL LRRC8A-MSP1E3D1. DCPIB was allowed to equilibrate and bind complex on ice for 1 hr prior to freezing grids. Sample with drug was cleared by a 5 min 21,000 x g spin prior to grid making. For freezing grids, a 3 μl drop of protein was applied to freshly glow discharged Holey Carbon, 400 mesh R 1.2/1.3 gold grids (Quantifoil, Großlöbichau, Germany). A FEI Vitrobot Mark IV (ThermoFisher Scientific) was utilized with 22°C, 100% humidity, one blot force, and a 3 s blot time, before plunge freezing in liquid ethane. Grids were then clipped in autoloader cartridges (FEI, Hillsboro, Oregon) and shipped in a dry shipper for data collection. The MSP2N2 sample was frozen without drug at 0.8 mg/mL with the same conditions as the MSP1E3D1 grids.

For the digitonin sample, purified protein was shipped in a refrigerated container. The next day, for freezing, a 3 μL drop of protein was applied to a freshly glow discharged Holey Carbon, 400 mesh R 1.2/1.3 gold grid (Quantifoil). A FEI Vitrobot Mark IV (ThermoFisher Scientific) was utilized with 22°C, 100% humidity, one blot force, and a 5 s blot time, before plunge freezing in liquid ethane. Grids were then clipped and used for data collection.

## Cryo-EM data acquisition

For the digitonin-solubilized channels in 150 and 600 mM KCl, grids were transferred to an FEI Titan Krios cryo-EM operated at an acceleration voltage of 300 kV. Images were recorded in an automated fashion with SerialEM (*Mastronarde, 2005*) with a defocus range of −1.2 ~ −2.5 μm over 8 s as 40 subframes with a Gatan K2 direct electron detector in super-resolution mode with a super-resolution pixel size of 0.544 Å. The electron dose was 8 e⁻/pixel/s at the detector level and total accumulated dose was 54 e⁻/Å$^2$. For the digitonin-solublized channels in 70 mM KCl, grids were transferred to an FEI Titan Krios cryo-EM operated at an acceleration voltage of 300 kV. Images were recorded in an automated fashion with Leginon (*Suloway et al., 2005*) with a defocus range of −1.2 ~ −2.5 μm over 8 s as 40 subframes with a Gatan K2 direct electron detector in super-resolution mode with a super-resolution pixel size of 0.536 Å. The electron dose was 9 e⁻/pixel/s at the detector level and total accumulated dose was 55.6 e⁻/Å$^2$. For MSP2N2 nanodisc-reconstituted samples, grids were transferred to an FEI Titan Krios cryo-EM operated at an acceleration voltage of 300 kV. Images were recorded in an automated fashion with Leginon with a defocus range of −1.2 ~ −2.5 μm over 10 s as 40 subframes with a Gatan K2 direct electron detector in super-resolution mode with a super-resolution pixel size of 0.536 Å.

The MSP1E3D1 nanodisc-reconstituted samples were recorded in two sessions. For the first session, grids were transferred to an FEI Titan Krios cryo-EM operated at an acceleration voltage of 300 kV. Images were recorded in an automated fashion with SerialEM with a defocus range of −1.2 ~ −2.5 μm over 8 s as 40 subframes with a Gatan K2 direct electron detector in super-resolution mode with a super-resolution pixel size of 0.544 Å. The electron dose was 9 e⁻/pixel/s at the detector level and total accumulated dose was 60.8 e⁻/Å$^2$. For the second session, grids were transferred to an FEI Titan Krios cryo-EM operated at an acceleration voltage of 300 kV. Images were recorded in an automated fashion with Leginon with a defocus range of −1.2 ~ −2.5 μm over 8 s as 40 subframes with a Gatan K2 direct electron detector in super-resolution mode with a super-resolution pixel size of 0.536 Å. The electron dose was 9 e⁻/pixel/s at the detector level and total accumulated dose was 55.6 e⁻/Å$^2$. Also see *Tables 1* and *2*.

## Cryo-EM data processing

Processing was carried out using Relion 3.0 (*Zivanov et al., 2019*; *Zivanov et al., 2018*). Movies were gain and motion corrected with the Relion MotionCor2 package (standalone MotionCor2 for the digitonin datasets) (*Zheng et al., 2017*), and the data were binned to 1.088 Å/pixel. Ctf estimation was performed with Gctf 1.06. For particle picking, 1000–2000 particles were picked manually to generate references for autopicking. For the MSP1E3D1 and MSP2N2 Relion-3 processed datasets, 2x particle binning was performed at extraction for initial particle cleanup, before re-extraction to 1.088 Å/pixel for final classification and refinement. For the other datasets, particles were 2x binned for all processing steps except final 2D comparisons.

For the contracted and expanded LRRC8A states, we first noted differences in the linker region during initial full particle classing and blurred helical density in this region during full particle refinement. We therefore used a mask encompassing the linker region and the bottom of the transmembrane helices during classification without angular sampling to separate particles in these two classes. To obtain high-resolution particles for the final reconstruction, masking out the LRR region was crucial. For the final particle sets, we also performed unmasked refinement, which generated the full particle map shown in *Figure 1* for the contracted state. The full particles for the expanded state were consistently more difficult to refine to high resolution and also saw no benefit from Ctf Refinement, potentially due to their additional LRR heterogeneity.

For symmetry testing on the digitonin, MSP2N2, and MSP1E3D1 datasets, 2x binned particles (2.176 Å/pixel or 2.298 Å/pixel for MSP2N2) were first cleaned using 2D classification and 3D Classification using C1 symmetry with the same Gaussian filtered initial reference. For final symmetry

analysis, three 3D Classifications were performed using the same reference and C1, C3, and C6 symmetry operations. For further comparison of digitonin and MSP2N2 particles, the best C1 classes were then used for final refinements in C3 and C6.

Overall resolution was estimated using Relion 3.0 and Phenix.mtriage. Local resolution was calculated using Relion. For a detailed pipeline see *Figure 1—figure supplements 3–10*.

## Modeling and refinement

Cryo-EM maps were sharpened using Phenix.autosharpen. The structures were modeled ab inito in Coot for all regions outside of the LRRs and refined in real space using Phenix.real_space_refine implementing Ramachandran and NCS restraints. Restraints for DCPIB and POPC ligands were generated using Phenix.elbow from SMILES string inputs and optimized with the eLBOW AM1 QM method. Validation tools in Phenix, EMRinger (*Barad et al., 2015*), and Molprobity were used to guide two subsequent rounds of iterative manual adjustment in Coot and refinement in Phenix. For cross-validation, atoms in the final model were randomly displaced up to 0.5 Å and refined against one half-map ('work'). FSC curves were then calculated between the refined model and each half-map ('work' and 'free') using Phenix.mtriage. The absence of significant differences between the FSC curves is indicates the model was not overfit to the original map. Superpositions with published LRRC8A-detergent structures, which were not used as guides during model building, also demonstrate good overall correspondence aside from symmetry and conformational changes. For illustration of average LRR position in *Figure 1*, the 1.8 Å crystal structure from PDB 6FNW was docked as a rigid body into unmasked maps using Phenix. Channel cavity measurements were made with HOLE implemented in Coot. Electrostatic potential was calculated using APBS-PDB2PQR (*Dolinsky et al., 2004*). Figures were prepared using PyMOL, Chimera, ChimeraX, Fiji, Prism, and Adobe Photoshop and Illustrator software.

## Acknowledgements

We thank the members of the Brohawn laboratory for support, input, and critical reading of the manuscript. We gratefully acknowledge Team Brohawn (B Phillips, H Falahati, J Quiroz) and R Sepela for help with recordings at the 2018 Electrophysiology course at the Marine Biological Institute. We thank E Montabana for training in grid preparation and sample screening and D Toso and P Tobias at the Berkeley Bay Area Cryo-EM facility for help with collection of preliminary data. We thank M de la Cruz at the Memorial Sloan Kettering Cancer Center Cryo-EM Facility and the staff at the Simons Electron Microscopy Center for help with data collection. SGB is a New York Stem Cell Foundation-Robertson Neuroscience Investigator. This work is supported by the New York Stem Cell Foundation (SGB), NIGMS grant DP2GM123496-01 (SGB), a McKnight Foundation Scholar Award (SGB), a Klingenstein-Simons Foundation Fellowship Award (SGB), an NIGMS postdoctoral fellowship F32GM128263 (DMK), NIH-NCI Cancer Center Support Grant (PO CA008748), a Josie Robertson Investigators award (RKH), and the Searle Scholars Program (RKH). Some of this work was performed at the Simons Electron Microscopy Center and National Resource for Automated Molecular Microscopy located at the New York Structural Biology Center, supported by grants from the Simons Foundation (SF349247), NYSTAR, and the NIH National Institute of General Medical Sciences (GM103310) with additional support from Agouron Institute (F00316) and NIH (OD019994).

## Additional information

### Funding

| Funder | Grant reference number | Author |
| --- | --- | --- |
| New York Stem Cell Foundation | NYSCF-R-N145 | Stephen G Brohawn |
| National Institute of General Medical Sciences | DP2GM123496-01 | Stephen G Brohawn |
| Klingenstein Third Generation Foundation | | Stephen G Brohawn |

Neuroscience | Structural Biology and Molecular Biophysics

| McKnight Endowment Fund for Neuroscience | | Stephen G Brohawn |
|---|---|---|
| National Cancer Institute | PO CA008748 | Richard K Hite |
| Searle Scholars Program | | Richard K Hite |
| Robertson Foundation | | Richard K Hite |
| National Institute of General Medical Sciences | F32GM128263 | David M Kern |

The funders had no role in study design, data collection and interpretation, or the decision to submit the work for publication.

## Author contributions

David M Kern, Conceptualization, Formal analysis, Validation, Investigation, Methodology, Writing—original draft, Writing—review and editing; SeCheol Oh, Investigation, Writing—review and editing; Richard K Hite, Resources, Formal analysis, Supervision, Funding acquisition, Validation, Project administration, Writing—review and editing; Stephen G Brohawn, Conceptualization, Resources, Formal analysis, Supervision, Funding acquisition, Validation, Investigation, Writing—original draft, Project administration, Writing—review and editing

## Author ORCIDs

David M Kern http://orcid.org/0000-0001-8529-9045
SeCheol Oh http://orcid.org/0000-0002-1685-5922
Richard K Hite http://orcid.org/0000-0003-0496-0669
Stephen G Brohawn http://orcid.org/0000-0001-6768-3406

## Decision letter and Author response

Decision letter https://doi.org/10.7554/eLife.42636.sa1
Author response https://doi.org/10.7554/eLife.42636.sa2

# Additional files

## Supplementary files

• Transparent reporting form

## Data availability

Final maps of LRRC8A-DCPIB in MSPE3D1 nanodiscs have been deposited to the Electron Microscopy Data Bank under accession codes EMDB-0562 (masked constricted state), and EMDB-0563 (masked expanded state) and atomic coordinates have been deposited in the PDB under IDs 6NZW (constricted state) and 6NZZ (expanded state). The original micrograph movies have been deposited to EMPIAR under accession codes EMPIAR-10258 and EMPIAR-10259. The map of apo-LRRC8A in MSP2N2 nanodiscs in a constricted state has been deposited with EMDB accession code EMDB-0564 and coordinates deposited in the PDB with ID 6O00.

The following datasets were generated:

| Author(s) | Year | Dataset title | Dataset URL | Database and Identifier |
|---|---|---|---|---|
| Kern DM, Oh S, Hite RK, Brohawn SG | 2019 | Atomic coordinates (apo-LRRC8A in MSP2N2 nanodiscs constricted state) | https://www.rcsb.org/structure/6O00 | Protein Data Bank, 6O00 |
| Kern DM, Oh S, Hite RK | 2019 | Final map of LRRC8A-DCPIB in MSP1E3D1 nanodiscs (masked constricted state) | https://www.ebi.ac.uk/pdbe/entry/emdb/EMD-0562 | Electron Microscopy Data Bank, EMDB-0562 |
| Kern DM, Oh S, Hite RK, Brohawn SG | 2019 | Final map of LRRC8A-DCPIB in MSP1E3D1 nanodiscs (masked expanded state) | https://www.ebi.ac.uk/pdbe/entry/emdb/EMD-0563 | Electron Microscopy Data Bank, EMDB-0563 |

| Kern DM, Oh S, Hite RK, Brohawn SG | 2019 | Atomic coordinates (LRRC8A-DCPIB in MSP1E3D1 nanodiscs constricted state) | https://www.rcsb.org/structure/6NZW | Protein Data Bank, 6NZW |
|---|---|---|---|---|
| Kern DM, Oh S, Hite RK, Brohawn SG | 2019 | Atomic coordinates (LRRC8A-DCPIB in MSP1E3D1 nanodiscs expanded state) | https://www.rcsb.org/structure/6NZZ | Protein Data Bank, 6NZZ |
| Kern DM | 2019 | Final map of apo-LRRC8A in MSP2N2 nanodiscs (masked constricted state) | https://www.ebi.ac.uk/pdbe/entry/emdb/EMD-0564 | Electron Microscopy Data Bank, EMDB-0564 |
| David M Kern, Se-Cheol Oh, Richard K Hite, Stephen G Brohawn | 2019 | Original micrograph movies | https://www.ebi.ac.uk/pdbe/emdb/empiar/entry/10258/ | Electron Microscopy Public Image Archive, EMPIAR-10258 |
| David M Kern, Se-Cheol Oh, Richard K Hite, Stephen G Brohawn | 2019 | Original micrograph movies | https://www.ebi.ac.uk/pdbe/emdb/empiar/entry/10259/ | Electron Microscopy Public Image Archive, EMPIAR-10259 |

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
