## [Decision Letter]

Thank you for submitting your article "Cryo-EM structures of the DCPIB-inhibited volume-regulated anion channel LRRC8A in lipid nanodiscs" for consideration by *eLife*. Your article has been reviewed by three peer reviewers, and the evaluation has been overseen by Kenton Swartz as the Reviewing Editor and Richard Aldrich as the Senior Editor. The following individual involved in review of your submission have agreed to reveal his identity: Roderick MacKinnon (Reviewer #3).

The reviewers have discussed the reviews with one another and the Reviewing Editor has drafted this decision to help you prepare a revised submission.

Summary:

In this manuscript Kern and colleagues use cryo-EM to determine the first inhibited structure of the volume regulated anion channel LRRC8A in lipid nanodiscs. These structures reveal two novel features of these channels: that the DCPIB inhibitor binds to the extracellular selectivity filter to sterically block permeation, and two novel conformations of the channel where the intracellular pore and linker are at different levels of constriction. The LCRR8A nanodisc structure presented here shows that the channel has 6-fold symmetric transmembrane region with heterogeneous LRRs rather than a trimer of dimers as seen in the detergent-bound structures. The authors propose that this different arrangement is due to the presence of lipids and to the different interactions of the protein with the environment. Interestingly, despite the conformational heterogeneity of the LRRs, the authors show that the linkers connecting the TM and these cytosolic domains adopt two distinct conformations suggesting a pathway for the allosteric activation of these channels in response to low salt conditions. Overall the manuscript is well written, the data robust and clearly described. The structures provide novel insights into the structural organization of the LRRC8A channels. Some of the description and interpretation needs further clarification.

Essential revisions:

1) The separation of the constricted and expanded states is a major conclusion in this work. 3D classification with symmetry can potentially average away the true conformational differences and lead to wrong conclusion. Therefore, we would like to see 3D classification without symmetry also reach the same conclusion. Specifically, in Figure 1—figure supplement 4, the last classification "Class3D with Bottom TM and linker mask" can be done using an alternative strategy. After refinement of 129k particles into one volume with C6 symmetry (like the 3.7 Å map above), the particles can be symmetry expanded to 780k, followed by no alignment 3D classification (8-12 classes) without symmetry and with the current mask.

2) Similarly, when drawing the important conclusion about the symmetry difference between the reconstructions of LRRC8A-1E3D1, LRRC8A in digitonin and LRRC8A-2N2 (Figure 4—figure supplement 2), we would also suggest first refining all good particles into one 3D reconstruction with C6 symmetry, symmetry expansion and then 3D classification without symmetry or alignment. In this way, the major conformers will appear with its original symmetry features.

3) General question: what is the physiological significance of a channel that opens in 70 mM electrolyte? In several places the structure is discussed in the context of this property. Under what circumstances does this condition arise?

4) Subsection “Structure of LRRC8A in lipid nanodiscs” third paragraph, the first paragraph of subsection “Constricted and expanded structures suggest a mechanism to relay conformational changes between an ionic strength sensor and channel gate” and Figure 3: The two structures are different but the change in pore diameter is small compared to the diameter of the pore in the constricted state. Do you think the structure represents an initial step in a larger conformational change? What (data) makes you think this is a functionally important structural difference?

5) Second paragraph of subsection “DCPIB inhibits LRRC8A through a cork-in-bottle mechanism”: Please clarify why the drug-free and drug-bound conditions (digitonin and nanodisc, respectively) are different. Do you have any data comparing structures under the same conditions? The fourth paragraph of subsection “Channel symmetry and heterogeneity of LRRs in nanodiscs” seems to suggest that you have drug-free structures in a MSP2N2 nanodisc. What does this show? We ask because sometimes density in the pore of an ion channel can be observed in some detergent/lipid systems and not in others. If structural data is available for the protein in nanodiscs without drug it would seem very important to include it in the present manuscript.

6) Do mutational data (perhaps in the literature) support the observed drug binding site? It would be useful to let the reader know this while describing the chemical interactions and mechanism of block (second and third paragraphs of subsection “DCPIB inhibits LRRC8A through a cork-in-bottle mechanism”) since the 6-fold axis has prevented you from being able to build the inhibitor into the density with much accuracy. If mutation experiments have not been carried out, we request that you include them to strengthen our confidence in the inhibitor model. It would also be interesting to consider undertaking ion competition experiments to test the proposed plugging mechanism.

7) Results section: "Movement of cytoplasmic regions, perhaps as a consequence reduced ionic strength, can thus be coupled to the N-terminal gating region and an expansion of the channel pore in a concerted fashion". This is a fairly vague statement. What precisely do you mean by this? Please be more explicit. What does the structure tell you about ionic strength sensing? Do you think that high ionic strength solutions screen charges, permitting them to get closer, for example?

8) Please clarify what you mean by the statement: "The relatively modest (low μM) affinity of DCPIB for LRRC8 channels (Decher et al., 2001) is probably due to the small protein-DCPIB interface and lack of significant structural rearrangement in the selectivity filter upon binding." The small interface would favor low affinity. The lack of required rearrangement would favor high. What point are you trying to make?

9) While enticing, the observed conformations do not provide a clear mechanistic link between the sensor for ionic strength and the observed conformational changes. Do the authors have any data to suggest that low salt favors the expanded conformation over the constricted one, or data showing how mutating residues in the linker region affect channel activation?

10) The implications of the different structural arrangement of the TM and cytosolic domains are not explored. Do the authors have any data addressing the role of the interactions of lipids with the intersubunit interfaces, for example, by testing how different lipid compositions affect channel function and structure?

11) The lack of conformational changes near the extracellular region between the constricted and expanded 3D reconstructions can be due to DCPIB binding. This possibility should be discussed.

---

## [Author Response]

Essential revisions:1) The separation of the constricted and expanded states is a major conclusion in this work. 3D classification with symmetry can potentially average away the true conformational differences and lead to wrong conclusion. Therefore, we would like to see 3D classification without symmetry also reach the same conclusion. Specifically, in Figure 1—figure supplement 4, the last classification "Class3D with Bottom TM and linker mask" can be done using an alternative strategy. After refinement of 129k particles into one volume with C6 symmetry (like the 3.7 Å map above), the particles can be symmetry expanded to 780k, followed by no alignment 3D classification (8-12 classes) without symmetry and with the current mask.

We thank the reviewers for their suggestion. As requested, we performed masked 3D classification without symmetry following C6 refinement and symmetry expansion (Figure 4—figure supplement 3A). This procedure leads to the same conclusion that the particles can be separated into constricted and expanded states. Both states show approximate six-fold symmetry in the linker, transmembrane, and extracellular regions of the channel. Weaker density for one or two subunits within some classes generated from the 129K particle set (of which 60K contribute to the final reconstructions, see Figure 1—figure supplement 5) suggests that a small fraction of the particles may adopt asymmetric conformations. However, we have seen similar differences in the relative strength of subunits of asymmetric 3D classes of other symmetric protein complexes and the differences may instead be due to particle misalignment, low signal-to-noise particle images or extreme disorder of the LRR domains biasing the particle alignments. To further increase our confidence in the classifications, we also performed refinement of the final set of particles without symmetry imposed. This asymmetric refinement generated approximately six-fold symmetric constricted and expanded states consistent with our conclusions (Figure 4—figure supplement 3B).

2) Similarly, when drawing the important conclusion about the symmetry difference between the reconstructions of LRRC8A-1E3D1, LRRC8A in digitonin and LRRC8A-2N2 (Figure 4—figure supplement 2), we would also suggest first refining all good particles into one 3D reconstruction with C6 symmetry, symmetry expansion and then 3D classification without symmetry or alignment. In this way, the major conformers will appear with its original symmetry features.

We performed this refinement with the LRRC8A-MSP2N2 and digitonin datasets and the results support the conclusions about symmetry differences we report (Figure 4—figure supplement 3A). We also refer to three reports of higher resolution reconstructions of LRRC8A in digitonin (Deneka et al., 2018; Kasuya et al., 2018; Kefauver et al., 2018a), which all reach similar conclusions to those presented here concerning the C3 symmetry of LRRC8A in detergent. Please see response to essential revision #5 for details on additional analyses of the LRRC8A-MSP2N2 dataset included in this revision.

3) General question: what is the physiological significance of a channel that opens in 70 mM electrolyte? In several places the structure is discussed in the context of this property. Under what circumstances does this condition arise?

We used 70 mM electrolyte in single channel recordings to robustly activate LRRC8A currents as has been reported in several other recent reports (Kasuya et al., 2018; Syeda et al., 2016; Deneka et al., 2018; Voets et al., 1999). Less dramatic reductions in ionic strength result in a lesser degree of activation in cells (Deneka et al., 2018). It is unknown what range of intracellular ionic strengths are physiologically relevant stimuli for this channel or generally under what circumstances these conditions arise, but experimental and modeling results are consistent with reduced intracellular ionic strength in response to hypoosmotic extracellular conditions activating LRRC8 channels (Voets et al., 1999).

4) Subsection “Structure of LRRC8A in lipid nanodiscs” third paragraph, the first paragraph of subsection “Constricted and expanded structures suggest a mechanism to relay conformational changes between an ionic strength sensor and channel gate” and Figure 3: The two structures are different but the change in pore diameter is small compared to the diameter of the pore in the constricted state. Do you think the structure represents an initial step in a larger conformational change? What (data) makes you think this is a functionally important structural difference?

We are unable at this time to determine whether the conformational changes we observe are sufficient to gate the channel open or if they represent an initial step of a larger conformational change to open the channel. The reason for this is that the pore dimensions of the channel that we calculate do not account for volume occupied by the N-terminus (amino acids 1-14) of each subunit. These residues from all six subunits project into the pore and may form the channel gate, but are not resolved in our structures (or other structures of LRRC8A to date). The true pore diameter is therefore smaller by an unknown amount in both the constricted and expanded states. While the expansion we observe is potentially large enough to open a pore, experiments to resolve the conformation of the N-terminus in each state are required to determine whether it is sufficient to gate the channel.

5) Second paragraph of subsection “DCPIB inhibits LRRC8A through a cork-in-bottle mechanism”: Please clarify why the drug-free and drug-bound conditions (digitonin and nanodisc, respectively) are different. Do you have any data comparing structures under the same conditions? The fourth paragraph of subsection “Channel symmetry and heterogeneity of LRRs in nanodiscs” seems to suggest that you have drug-free structures in a MSP2N2 nanodisc. What does this show? We ask because sometimes density in the pore of an ion channel can be observed in some detergent/lipid systems and not in others. If structural data is available for the protein in nanodiscs without drug it would seem very important to include it in the present manuscript.

We agree with these points. In order to distinguish differences that are a consequence of drug binding from differences that are a consequence of different hydrophobic environments, we have improved our refinement of the LRRC8A-MSP2N2 without DCPIB dataset (by incorporating per-particle motion trajectories and CTF refinement) and include classification results and a reconstruction to 4.18 Å resolution in the revised manuscript (Figure 1—figure supplements 8-10). The results are consistent with our conclusions and are summarized as follows. Drug binding results in density modeled as DCPIB in the selectivity filter and an increase in the conformational space sampled by LRRs. DCPIB density is only observed in reconstructions of DCPIB-bound LRRC8A in nanodiscs, while reconstructions of apo-LRRC8A in nanodiscs and apo-LRRC8A in digitonin both lack this density. Three additional differences, unrelated to drug binding, are found between structures of LRRC8A determined in lipid nanodiscs and digitonin detergent. First, large scale differences in structural organization are observed. In lipid nanodiscs, LRRC8A is six-fold symmetric in the extracellular, transmembrane, and linker regions with asymmetric and structurally heterogeneous LRRs, while in digitonin LRRC8A is three-fold symmetric with compact LRRs. Second, in lipid nanodiscs, but not detergent, density consistent with lipid binding between subunit interfaces is observed. Third, in lipid nanodiscs, but not detergent, particles are distributed between well-resolved constricted and expanded states in roughly similar proportions. We hypothesize that the bound lipids in nanodiscs structures, which are apparently lost during detergent solubilization prior to structure determination in digitonin, account for these differences between nanodisc-embedded and detergent-solubilized structures.

6) Do mutational data (perhaps in the literature) support the observed drug binding site? It would be useful to let the reader know this while describing the chemical interactions and mechanism of block (second and third paragraphs of subsection “DCPIB inhibits LRRC8A through a cork-in-bottle mechanism”) since the 6-fold axis has prevented you from being able to build the inhibitor into the density with much accuracy. If mutation experiments have not been carried out, we request that you include them to strengthen our confidence in the inhibitor model. It would also be interesting to consider undertaking ion competition experiments to test the proposed plugging mechanism.

Yes. We have included a reference to mutational data reported by the Patapoutian and Ward groups (Kefauver et al., 2018) supporting the observed drug binding site. Mutation of the arginine in the extracellular selectivity filter of LRRC8A to phenylalanine (R103F) reduces the extent of extracellular ATP block of LRRC8AC heteromeric channels from 72% ± 2% to 2 ± 3%. The R103F mutation creates an apolar selectivity filter ring in these heteromeric channels because the residue corresponding to R103 in LRRC8A is a leucine in LRRC8C. This is consistent with our model of electrostatic interactions being an important determinant for binding of anionic blockers including DCPIB and ATP to this site. We agree that our model makes interesting predictions that could be tested by ion competition experiments, but prefer to include these in future work.

7) Results section: "Movement of cytoplasmic regions, perhaps as a consequence reduced ionic strength, can thus be coupled to the N-terminal gating region and an expansion of the channel pore in a concerted fashion". This is a fairly vague statement. What precisely do you mean by this? Please be more explicit. What does the structure tell you about ionic strength sensing? Do you think that high ionic strength solutions screen charges, permitting them to get closer, for example?

We have clarified this idea to be more explicit in proposing possible mechanisms including charge screening and moved the sentence to the discussion. We have also emphasized that while the ionic strength sensor has been presumed to be an intracellular aspect of the channel, it is not known which residue(s) are involved and we can only speculate on possible mechanisms since the cytoplasmic LRRs and long cytoplasmic linkers are heterogenous and poorly resolved in our structures.

8) Please clarify what you mean by the statement: "The relatively modest (low μM) affinity of DCPIB for LRRC8 channels (Decher et al., 2001) is probably due to the small protein-DCPIB interface and lack of significant structural rearrangement in the selectivity filter upon binding." The small interface would favor low affinity. The lack of required rearrangement would favor high. What point are you trying to make?

We thank the reviewers for pointing out this editing mistake. The point was that despite a rigid binding site, the interface is small and binding is therefore weak. We add that development of higher affinity and subunit selective blockers might be achieved by increasing the size and complementarity of the interaction surface.

9) While enticing, the observed conformations do not provide a clear mechanistic link between the sensor for ionic strength and the observed conformational changes. Do the authors have any data to suggest that low salt favors the expanded conformation over the constricted one, or data showing how mutating residues in the linker region affect channel activation?

We agree with the reviewers and do not have these data, so we are careful to point out this out in the Discussion.

10) The implications of the different structural arrangement of the TM and cytosolic domains are not explored. Do the authors have any data addressing the role of the interactions of lipids with the intersubunit interfaces, for example, by testing how different lipid compositions affect channel function and structure?

The major differences that we report are a result of varied hydrophobic environments between the structures reported here in lipid nanodiscs and those reported previously in digitonin. Since we are not able to assay channel function in detergent, we are unable to assess the functional implications of these differences at this time. We agree the effect of different lipid compositions on channel function will be interesting to pursue. We prefer to include it in later, systematic study. One reason for this is that the functional consequences of varying lipid compositions are difficult to predict from our structures because the lipid interactions observed are between lipid hydrocarbon tails and hydrophobic surfaces of the channel.

11) The lack of conformational changes near the extracellular region between the constricted and expanded 3D reconstructions can be due to DCPIB binding. This possibility should be discussed.

We conclude that the conformation near the extracellular region in the constricted state is not strongly influenced by drug binding, since the constricted structures in MSPE3D1 with DCPIB and MSP2N2 without DCPIB are indistinguishable in this region (Figure 1—figure supplement 10D). The structure of this region is also very similar in apo-LRRC8A structures in digitonin. However, we cannot rule out that the structure of the extracellular region in the expanded state is influenced by drug binding as we were not able to resolve an expanded structure in MSP2N2 nanodiscs without DCPIB to high resolution.